# Effects of Dietary Riboflavin Supplementation on the Growth Performance, Body Composition and Anti-Oxidative Capacity of Coho Salmon (*Oncorhynchus kisutch*) Post-Smolts

**DOI:** 10.3390/ani12223218

**Published:** 2022-11-20

**Authors:** Hai-Rui Yu, Meng-Jie Guo, Le-Yong Yu, Ling-Yao Li, Qing-He Wang, Fang-Hui Li, Yu-Zhuo Zhang, Jia-Yi Zhang, Ji-Yun Hou

**Affiliations:** 1Key Laboratory of Biochemistry and Molecular Biology in Universities of Shandong (Weifang University), Weifang Key Laboratory of Coho Salmon Culturing Facility Engineering, Institute of Modern Facility Fisheries, College of Biology and Oceanography, Weifang University, Weifang 261061, China; 2Guangxi Key Laboratory for Polysaccharide Materials and Modifications, Key Laboratory of Protection and Utilization of Marine Resources, Guangxi Minzu University, Nanning 530008, China; 3Shandong Collaborative Innovation Center of Coho Salmon Health Culture Engineering Technology, Shandong Conqueren Marine Technology Co., Ltd., Weifang 261108, China; 4Tianjin Key Lab of Aqua-Ecology and Aquaculture, College of Fisheries, Tianjin Agricultural University, Tianjin 300384, China

**Keywords:** antioxidant, body composition, *Oncorhynchus kisutch*, growth performance, riboflavin

## Abstract

**Simple Summary:**

In China, with the improvement of living standards, consumer demand for coho salmon is growing rapidly. As the only source of flavin nucleotides, exogenous riboflavin is an indispensable nutrient to promote the growth of fish and regulate physiological functions. Therefore, it is particularly relevant to full understanding of the dietary riboflavin requirements of coho salmon. The physiological function of riboflavin is mainly as coenzyme FMN and FAD and covalently bound flavin precursor, involved in the body’s macromolecular metabolism and redox reaction. However, there is incognizance of riboflavin requirements in this fish species. In this study, seven isoenergetic experimental diets containing different riboflavin levels were formulated to study the changes in growth performance, body composition and antioxidant capacity of coho salmon. The results showed that different dietary riboflavin levels affected the growth performance, body composition and antioxidant capacity of coho salmon post-smolts. The optimum riboflavin requirement was determined according to the specific growth rate (SGR).

**Abstract:**

The present study investigated the effects of dietary riboflavin on growth performance, body composition and anti-oxidative capacity of coho salmon (*Oncorhynchus kisutch*) post-smolts. Seven experimental diets were formulated with graded riboflavin levels of 0.00, 3.96, 8.07, 16.11, 31.81, 63.67 and 126.69 mg/kg, respectively. Each diet was fed to triplicate groups of 10 fish with an individually initial mean body weight of 186.22 ± 0.41 g in 21 cages (water volume, 1000-L/cage) and fed three times daily (7:30, 12:30 and 17:30) to apparent satiation for 12 weeks. Fish fed a diet with 31.81 mg/kg riboflavin had the highest specific growth rate (SGR), which was significantly higher than fish-fed diets with 0.00, 3.96, 8.07 and 126.69 mg/kg riboflavin (*p* < 0.05). Feed conversion ratio showed an inverse trend with SGR. No significant differences were observed in condition factor, hepatosomatic index, viscerosomatic index, muscle moisture, crude protein and ash contents among dietary groups. Muscle lipid had the highest content in the 31.81 mg/kg group and was significantly higher (*p* < 0.05) than those in the 0.00, 3.96 and 8.07 mg/kg groups. The alanine aminotransferase, aspartate aminotransferase and malondialdehyde contents in the liver and serum of fish were significantly decreased with the increase in dietary riboflavin level up to 31.81 mg/kg, and then increased as dietary riboflavin level further increased. An inverse trend was observed for total superoxide dismutase and catalase activities. Serum total cholesterol and triglyceride levels were significantly decreased with the dietary of riboflavin levels up to 31.81 and 63.67 mg/kg, respectively. The cubic curve regression analysis based on SGR indicated that the optimum dietary riboflavin level was estimated to be 35.26 mg/kg for coho salmon post-smolts.

## 1. Introduction

As one of the group B vitamins, riboflavin binds to specific enzyme proteins with adenine dinucleotide and mononucleotide as cofactors in vivo to form various flavin proteases. These enzymes are closely related to the metabolism of macromolecules in the body and play crucial roles in the transfer of hydrogen atoms during biological oxidation [1,2]. In people and animals, the only source of flavin nucleotides is exogenous riboflavin, which can be synthesized by plants, yeasts, fungi and other microorganisms [3]. Animals require riboflavin for the growth and repair of tissues. Riboflavin deficiency affects iron absorption, mitochondrial dysfunction, gastrointestinal dysfunction, brain dysfunction, membrane angiogenesis and indirect antioxidant defenses in vivo [4,5]. In farmed fish species, such as Bloch (*Channa punctatus*), grass carp (*Ctenopharyngodon idella*), Jian carp (*Cyprinus carpio var*. Jian), rainbow trout (*Oncorhynchus mykiss*) and tilapia (*Oreochromis mossambicus × O. niloticus*), dietary riboflavin supplementation has a positive impact on growth, carcass composition, antioxidant defense as well as intestinal enzyme activities [6,7,8,9,10]. Riboflavin deficiency also led to almost complete cessation of growth while prolonged deficiency of riboflavin also caused eye clouding in the three trout species [11]. These studies have demonstrated that loss of appetite, growth retardation and increased mortality are common in fish with riboflavin deficiency.

In recent years, it has been found that riboflavin is crucial to the body’s antioxidant defense system [12]. Riboflavin may protect the body from reperfusion oxidative damage through its ability to scavenge free radicals [13]. Tripathi et al. [14] reported that riboflavin had neuroprotective effects against cerebral ischemia. Aerobic cells need to produce energy through oxidative phosphorylation, which is one of the reasons why riboflavin deficiency affects the growth and development of the organism [15]. Riboflavin has properties as an endogenous monoclinic oxygen sensitizer and its photochemical and enzymatic breakdown products are non-toxic [16]. It has been proven that the addition of dietary riboflavin improved the antioxidant status of ducks and enhanced their reproductive performance [17]. Leghorn chickens fed the riboflavin deficiency diet fail to produce functional riboflavin-binding protein, resulting in excessive hepatic lipid accumulation, impaired lipid β-oxidation and severe hypoglycemia before death [18]. Riboflavin effectively reduced lung damage caused by lipopolysaccharide and enhanced antioxidant capacity and antitumor activity in mitochondria of rats with breast cancer when administered in combination with triamcinolone [19,20]. The deficiency of riboflavin in juvenile grass carp led to sluggish growth, reduced feeding and a decline in immunity [21].

Coho salmon (*Oncorhynchus kisutch*) spawns and develops in freshwater and lives in the sea when grown up; its original habitat is mainly in the eastern Pacific Ocean. In recent years, coho salmon has become one of the most promising farmed salmon species in China because of its rapid growth, high economic value, rich nutrition and delicious meat quality. Until now, studies on nutritional requirements of vitamins for coho salmon are still deficient except for ascorbic acid [22] and vitamin E [23], and no information is available on dietary riboflavin requirement for this fish species. Hence, the present study evaluated the effects of graded dietary riboflavin levels on growth performance, body composition and the antioxidant response of coho salmon post-smolts.

## 2. Materials and Methods

### 2.1. Experimental Diets

The formulation and proximate composition of the diets were presented in Table 1. The purified basal diet was formulated using casein as a protein source and fish oil and corn oil as lipid sources. Seven experimental diets were formulated by supplementation with riboflavin (contains 95.63%, Hubei Guangji Pharmaceutical Co., Ltd., Wuxue, China) at the level of 0, 4, 8, 16, 32, 64 and 128 mg/kg diets. The dietary riboflavin contents were analyzed by high performance liquid chromatography (HPLC, HP1100, Agilent Technologies, Palo Alto, CA, USA) to contain 0.00, 3.96, 8.07, 16.11, 31.81, 63.67 and 126.69 mg/kg, respectively. The diets were prepared by screw extruding, air-drying and storing in aluminum foil bags at −20 °C until they were needed.

### 2.2. Feeding Trial

The post-smolts used in this trial were purchased from the National Center for Introduction and Breeding of Aquatic Products (Beijing, China) and cultured in one base of the Coho Salmon Health Culture Engineering Technology Center of Shandong Collaborative Innovation Center (Weifang, China). Briefly, 10 fish each weighing 186.22 ± 0.41 g were randomly distributed into each of the 21 floating cages (1 m × 1 m × 1 m, L × W × H, 1000 L water volume) in an earthen pond connected to a flow-through rearing system with continuous filtered underground fresh spring water. During the 12-week feeding period, fish were hand-fed at 3–5% of their body weight three times a day (7:30, 12:30, 17:30). Water temperature ranged from 14 to 16 °C, while the pH ranged between 7.1 and 7.5, and the dissolved oxygen levels surpassed 7.8 mg O_2_/L. Natural light regimes were used during the rearing of the post-smolts.

### 2.3. Sampling Procedures

At the end of the feeding trial, the fish were fasted for 24 h, anesthetized with tricaine methanesulfonate (MS-222) at a concentration of 30 mg/L, and counted and weighed in bulk to determine the survival rate and growth performance. Three fish from each replicate were tested for morphological indices including condition factor (CF), hepatosomatic index (HSI) and viscerosomatic index (VSI). Five additional fish were randomly sampled for collecting serum samples from their caudal veins with a disposable medical syringe and then stored at room temperature for 2 hrs. Using a centrifuge set to 3500× *g* for ten minutes at 4 °C, the serum samples were collected and stored at −80 °C for the evaluation of anti-oxidative factors and biochemical parameters. After being bled, the five fish were quickly dissected with their livers and muscles removed, which were stored at −80 °C. The liver samples were analyzed for antioxidative parameters, whereas muscle samples were analyzed for muscle proximate composition.

### 2.4. Analytical Methods

#### 2.4.1. Growth Performance

The indicator calculation formula is as follows:$$\mathrm{Survival}\;\mathrm{rate}\;(\mathrm{SR},\;\%)=100\times \frac{\left(\mathrm{final}\;\mathrm{amount}\;\mathrm{of}\;\mathrm{fish}\right)}{\left(\mathrm{inital}\;\mathrm{amount}\;\mathrm{of}\;\mathrm{fish}\right)}$$
$$\mathrm{Weight}\;\mathrm{growth}\;\mathrm{rate}\;(\mathrm{WGR},\;g)=100\times \frac{\mathrm{final}\;\mathrm{body}\;\mathrm{weight}-\mathrm{initial}\;\mathrm{body}\;\mathrm{weight}}{\mathrm{final}\;\mathrm{body}\;\mathrm{weight}}$$
$$\mathrm{Specific}\;\mathrm{growth}\;\mathrm{rate}\;(\mathrm{SGR},\;\%/\mathrm{day})=100\times \frac{\left[\mathrm{In}\;(\mathrm{final}\;\mathrm{body}\;\mathrm{weight})-\mathrm{In}\;(\mathrm{initial}\;\mathrm{body}\;\mathrm{weight})\right]}{\mathrm{days}}$$
$$\mathrm{Feed}\;\mathrm{conversion}\;\mathrm{ratio}\;(\mathrm{FCR})=\frac{\mathrm{total}\;\mathrm{feed}\;\mathrm{intake}}{\mathrm{final}\;\mathrm{body}\;\mathrm{weight}-\mathrm{initial}\;\mathrm{body}\;\mathrm{weight}}$$
$$\mathrm{Condition}\;\mathrm{factor}\;(\mathrm{CF},g/{\mathrm{cm}}^{3})=100\times \frac{\mathrm{body}\;\mathrm{weight}}{\mathrm{body}\;{\mathrm{length}}^{3}}$$
$$\mathrm{Hepatosomatic}\;\mathrm{index}\;(\mathrm{HSI},\%)=100\times \frac{\mathrm{liver}\;\mathrm{weight}}{\mathrm{body}\;\mathrm{weight}}$$
$$\mathrm{Viscerosomatic}\;\mathrm{index}\;(\mathrm{VSI},\%)=100\times \frac{\mathrm{viscera}\;\mathrm{weight}}{\mathrm{body}\;\mathrm{weight}}$$

#### 2.4.2. Proximate Composition Analysis

Samples of the diets and fish were analyzed for crude protein (by Kjeldahl apparatus, nitrogen × 6.25), crude lipid (ether extraction), ash (heating at 550 °C for 24 h in a muffle furnace) and moisture (dried at 105 °C) [24]. The gross energy content was measured using an oxygen bomb calorimeter Parr 1281 (Parr, Moline, IL, USA).

#### 2.4.3. Biochemical Tissue Analysis

The homogenate samples were collected for analyses after homogenization in pH 7.4 Tris-HCl buffer at 4 °C. The malondialdehyde (MDA) content in serum and liver were determined at 532 nm by thiobarbituric acid (TBA) method (the reference number of the commercial kit was A003-1) [25]. Superoxide dismutase activity in liver was determined at 550 nm by xanthine oxidase method (SOD, A001-1) [25]. The hepatic catalase (CAT, A007-1-1) activity was determined at 405 nm by ammonium molybdate method [26]. Aspartate aminotransferase (AST, C010-2-1) and alanine aminotransferase (ALT, C009-2-1) activities in serum and liver were determined by the method of Reitman and Frankel [27]. Determination of serum total cholesterol content (A111-1-1) at 500 nm by cholesterol oxidase (COD-PAP) method and serum triglyceride (A110-1-1) was determined at 510 nm by glyceraldehyde phosphate oxidase (GPO-PAP) method [28]. All indexes were determined using commercial reagent kits (Nanjing Jiancheng Bioengineering Institute, Nanjing, China), all of these were measured using a microplate reader (Tecan Spark10M, Salzburg, Austria). The liver riboflavin contents were analyzed by high performance liquid chromatography (HPLC, HP1100, Agilent Technologies, Palo Alto, CA, USA).

### 2.5. Statistical Analyses

All data were presented as means ± standard deviations (SD), and the analyses were performed using SPSS version 25.0 (SPSS Inc., Chicago, IL, USA). To compare the means of individual treatments, the significance was determined using a one-way analysis of variance (ANOVA) followed by Tukey’s test, which was used to verify statistical significance. The significant difference was set at *p* < 0.05. A cubic curve regression analysis was used to determine the optimal level of dietary riboflavin.

## 3. Results

### 3.1. Growth Performance

SR, CF, HSI and VSI of fish did not differ significantly (*p* > 0.05) among all the groups (Table 2). Fish-fed diet with 31.81 mg/kg riboflavin had the highest SGR, which was significantly higher than fish fed diets with 0.00, 3.96, 8.07 and 126.69 mg/kg riboflavin (*p* < 0.05). An inverse trend was observed for FCR, which was the lowest in the group of 31.81 mg/kg. Analyzing the relationship between the dietary riboflavin level and SGR with a cubic curve regression model revealed that the optimal dietary riboflavin level was 35.26 mg/kg (Figure 1).

### 3.2. Muscle Proximate Composition

The dietary riboflavin levels had no significant (*p* > 0.05) effects on muscle proximate compositions except for crude lipid, which had the highest content in the 31.81 mg/kg group and significantly higher (*p* > 0.05) than those in 0.00, 3.96 and 8.07 mg/kg groups (Table 3). The muscle contents of moisture, crude protein and ash ranged between 74.25–75.67%, 19.85–20.39% and 2.32–2.68%, respectively.

The hepatic riboflavin contents were remarkably improved with increasing of dietary riboflavin level up to 16.11 mg/kg, and then plateaud. The stable value of liver riboflavin content was approximate to 13–14 mg/kg (Table 4).

### 3.3. MDA, T-SOD, CAT, ALT, AST Activities in Liver

The contents of MDA and ALT activity in liver were significantly decreased with increasing of dietary riboflavin level up to 31.81 mg/kg, and then significantly increased as dietary riboflavin level further increased (Figure 2A,D). An inverse trend was observed for T-SOD and CAT activities, which were the highest in dietary riboflavin level of 31.81 mg/kg (Figure 2B,C). The hepatic AST content decreased significantly with an increase in dietary riboflavin up to 63.67 mg/kg (*p* < 0.05), which did not differ significantly from the group of 31.81 mg/kg but was significantly lower than the group of 126.69 mg/kg (Figure 2E).

### 3.4. MDA, Triglyceride, Total Cholesterol, ALT, AST Activities in Serum

The activities of MDA, total cholesterol and AST in serum of coho salmon decreased significantly with the increase of dietary riboflavin level up to 31.81 mg/kg (*p* < 0.05), which was significantly lower than those of groups of 63.67 and 126.69 mg/kg (Figure 3A,C,E). The serum triglyceride content decreased significantly with the increase in dietary riboflavin level up to 63.67 mg/kg, which did not differ significantly from that of the 31.81 mg/kg group but was significantly lower than that of the 126.69 mg/kg group (Figure 3B). The serum ALT content decreased significantly with the increase of dietary riboflavin level up to 31.81 mg/kg and then leveled off (Figure 3D).

## 4. Discussion

### 4.1. Growth Performance in Coho Salmon

Riboflavin plays an important role in the growth and development of fish, and insufficient or excessive content of dietary riboflavin leads to poor growth, decreased appetite, low feed efficiency, high mortality and other hypovitaminosis including photophobia, cataracts and movement disorders in many fish species [1,11,29,30,31,32,33]. In the present study, the suitable supplementation of dietary riboflavin significantly increased the SGR of post-smolts, which was probably because of the role riboflavin-5 phosphate and flavin adenosine dinucleotide may have promoted the metabolism of protein, lipid and carbohydrate in the body, and then promoted the growth of fish [1]. The growth performance and FCR of post-smolts were poor when fed diets with low riboflavin levels (0.00 and 3.96 mg/kg); when dietary riboflavin was deficient, the fish obviously grew slowly, which can be observed from the weight gain rate, and other symptoms caused by riboflavin deficiency were not observed in post-smolts during the feeding trial. Similar results were also found in other fish species such as red hybrid tilapia fingerlings [8], channel catfish *Ictalurus punctatus* fingerlings [30] and juvenile sunshine bass *Morone chrysops × Morone saxatilis* [29]. According to the SGR of post-smolts, cubic curve regression analysis showed that post-smolts had the best growth when dietary riboflavin content was 35.26 mg/kg, which was similar to the results obtained in rainbow trout (5–15 mg/kg) [34] and Pacific salmon (20–25 mg/kg) [35]. In a study of Atlantic salmon, 6–8 mg/kg riboflavin was added to fishmeal-based diets to meet growth needs, whereas plant-based diets were far from sufficient [30]. The present study was based on the addition of riboflavin to purified feeds, so the demand for dietary riboflavin was higher. However, comparatively low dietary riboflavin requirements were reported in other fish species, such as Atlantic salmon (10–12 mg/kg) [36], juvenile grass carp (6.65 mg/kg) [37], Jian carp (5.0 mg/kg) [7] and sunshine bass (4.1–5.0 mg/kg) [29]. The probable reasons might be attributed to the different fish species and sizes used in different experiments. Meanwhile, the vitamin requirement may also be related to the nutritional level and quality of feed, the experimental conditions and so on.

In the present study, the CF, HSI and VSI did not differ significantly among the groups. As for the HSI, it reflects the health of the liver to some extent health degree, but the effect of riboflavin on the HSI had different results in different fish species. For example, the HSI of juvenile sunshine bass in the dietary riboflavin deficient group was significantly lower than those in other supplemental groups [29], while the opposite result was observed in gilthead seabream *Sparus aurata* L. [38]. Brønstad, Bjerkås and Waagbø [39] suggested that this difference might be related to the lipid level in experimental diets. However, there is still a lack of reports on the interaction between dietary lipid levels and riboflavin in fish nutrition studies, so the effect of riboflavin on the HSI of different fish species needs to be further studied.

### 4.2. Muscle Proximate Composition

Muscle nutrient content is an important indicator of fish quality [40]. Adding dietary riboflavin can improve the nutritional value and taste of fish meat [6] and dried fish products [41]. In the present study, the dietary riboflavin supplementation did not significantly affect the moisture, crude protein and ash contents of fish muscle; however, muscle crude lipid content was the highest at the group of 31.18 mg/kg. Similar results were also observed in salmonids [33] and European sea bass [42]. Previous studies have shown that the levels of crude protein and crude ash were not directly related to the diet composition but related to the age and size of fish [33,43]. The crude lipid level in muscle increased significantly with the dietary riboflavin level increasing from 0 mg/kg to 31.18 mg/kg and exhibited a relatively consistent trend when the dietary riboflavin level further increased. A similar result was also reported by Li et al. [7] in Jian carp. Dietary riboflavin supplementation reduced muscle lipid content in fish, which was also confirmed in Atlantic salmon [39]. Studies on duck breeds and duck embryos showed that riboflavin was related to lipid metabolism, whereas flavoproteins were important proteins involved in lipid metabolism, and lower activities of flavoproteins inhibited fatty acid oxidation [44]. In addition, the effect of riboflavin on lipid metabolism may be achieved by controlling the expression of apolipoprotein B100 [45]. The mechanism of riboflavin on lipid metabolism needs further study.

Maximum liver riboflavin storage can be used as a reference for riboflavin requirement [1]. The riboflavin storage in fish liver was significantly lower in 0 mg/kg riboflavin diet group, it was difficult to obtain riboflavin from purified feed. It was consistent with Hughes’ research on rainbow trout [46]. Studies have shown that dietary fish meal can meet the basic needs of fish growth in the 6–8 mg/kg diet [40]. The riboflavin storage in liver that reached 11.84 mg at a riboflavin level of 3.96 mg/kg did not inhibit the growth of fish [46]. There was no significant difference in liver riboflavin storage when riboflavin content was higher than 16.11; superfluous riboflavin was excreted, same as Woodward’s conclusion [9]. In rainbow trout, dietary riboflavin concentration of 6 mg/kg reached the maximum liver riboflavin storage [45]. Other studies have shown that riboflavin levels in the spleen, head kidney, and hind kidney are more sensitive to riboflavin gradient diets than in the liver [9].

### 4.3. Liver and Serum Biochemical Indicators

The organism produces oxygen free radicals in the process of respiration and metabolism, the body has different degrees of oxidative damage when oxygen free radicals are excessive [47,48] and timely removal of excess free radicals can reduce the degree of oxidative damage. SOD, CAT and other antioxidant enzymes are important enzymes to eliminate oxygen free radicals in the animal body and play an important role in the balance between oxidation and antioxidant [49]. The activities of SOD and CAT in the serum of Jian carp [7,50] could be significantly increased by the appropriate amount of riboflavin. In the present study, a similar result was found that appropriate riboflavin supplementation significantly increased the activities of CAT and SOD in the liver and serum of post-smolts. MDA content is an important parameter of the body’s potential antioxidant capacity, reflecting the rate and intensity of lipid peroxidation in the body. Riboflavin deficiency has been reported to reduce the antioxidant capacity and immunity of grass carp [37]. With the increase in dietary riboflavin supplemental level, MDA content in the serum of fingerling channel catfish showed a downward trend [51]. Jiang et al. [6] also found that elevated MDA content was observed in riboflavin-deficient treatment in grass carp. The riboflavin supplementation significantly decreased liver and serum MDA content of coho salmon post-smolts. These results revealed that the increase of riboflavin level could reduce the extent of the free radical attack and oxidative stress damage in fish to a certain extent [49], i.e., the appropriate supplementation of dietary riboflavin is helpful to improve the antioxidant capacity of fish. Perhaps it was because enough dietary riboflavin could improve the ability to resist superoxide anion and oxygen free radicals [21].

AST and ALT are important amino acid transaminases widely existing in the cytoplasm and mitochondria of fish that play a critical role in the process of protein metabolism in fish; they are commonly evaluated in fish that encounter toxins, stress, disease and/or malnutrition and reaction liver function [52]. Researchers demonstrated that riboflavin deficiency could cause damage to liver structure and function [53]. When dietary riboflavin content was low, liver AST and ALT were higher, riboflavin deficiency may have affected the metabolic balance of amino acids. Transaminase activity fluctuated at excessive riboflavin levels. Numerous previous studies have shown that feeding high concentrations of riboflavin has no negative effect on rainbow trout; superfluous riboflavin may not affect liver health [54]. High serum AST and ALT activities indicated liver damage [52]; it is also the case in the present study that the serum AST and ALT activities were decreased with increasing dietary riboflavin levels, indicating that the liver of post-smolts might be damaged when fed a diet with insufficient riboflavin concentration.

Serum triglyceride and total cholesterol decreased gradually with the increase in dietary riboflavin level, consistent with the results of Li et al. on Jian carp [7]. Riboflavin can inhibit lipid peroxidation and reduce blood lipids, and can reduce the contents of serum triglycerides, low-density protein cholesterol and total cholesterol [55]. This may be because riboflavin can promote the growth and development of fish intestine and increase the activity of fat metabolism enzymes, thereby enhancing its digestion of nutrients.

## 5. Conclusions

The present study showed that suitable dietary riboflavin level significantly improved the growth, FCR and activities of hepatic anti-oxidative enzymes (T-SOD and CAT); reduced the hepatic and serum MDA content, serum total cholesterol and triglyceride levels in coho salmon post-smolts. Based on SGR, the dietary riboflavin requirement was estimated to be 35.26 mg/kg for coho salmon post-smolts.

## Figures and Tables

**Figure 1 animals-12-03218-f001:**
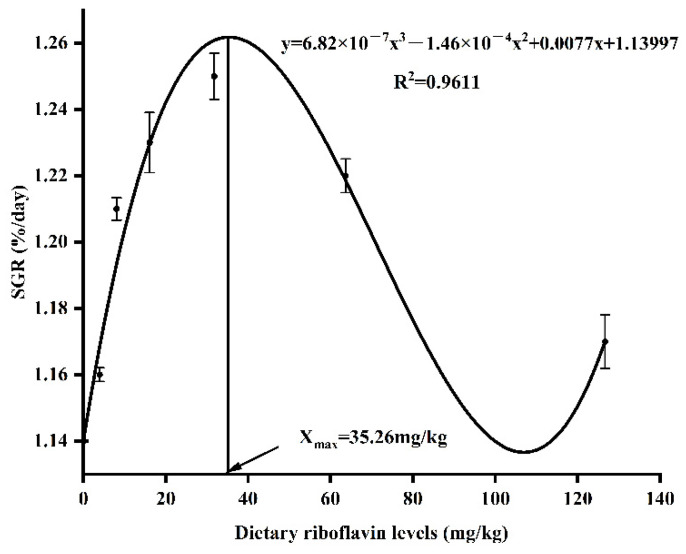
A cubic curve regression analysis of the relationship between specific growth rate (SGR) and dietary riboflavin level revealed that the optimal dietary riboflavin level was 35.26 mg/kg for coho salmon (*Oncorhynchus kisutch*) post-smolts.

**Figure 2 animals-12-03218-f002:**
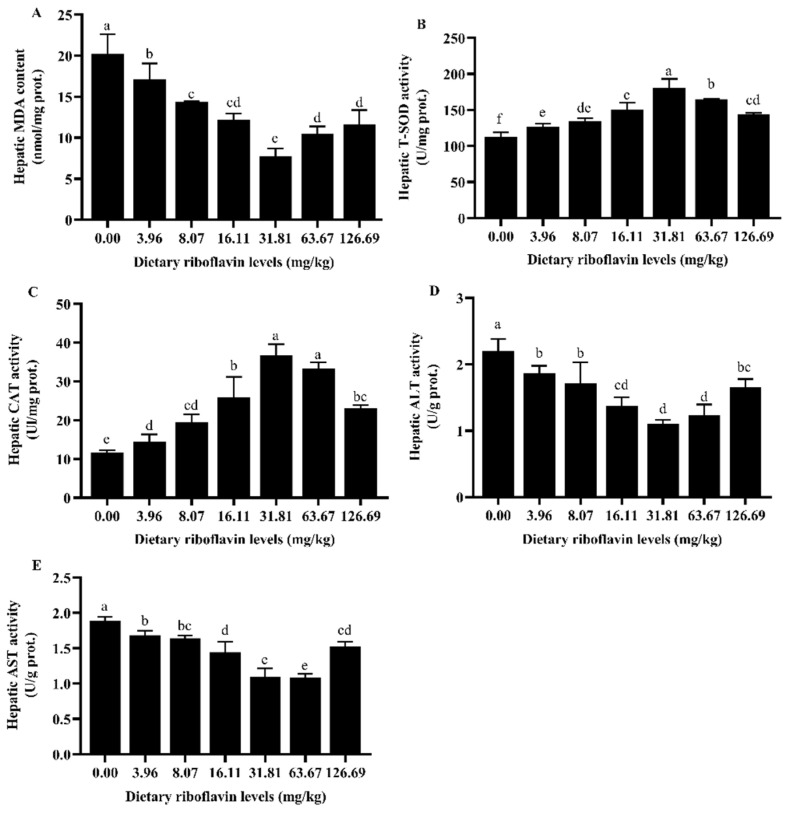
Effect of dietary riboflavin levels on hepatic MDA, T-SOD, CAT, ALT and AST activities of coho salmon (*Oncorhynchus kisutch*) post-smolts for 12 weeks. (**A**) MDA; (**B**) T-SOD; (**C**) CAT; (**D**) ALT; (**E**) AST. Values are expressed as mean ± SD of three replicate groups. Means with different superscript letters are significantly different (*p* < 0.05). Abbreviations: MDA, malondialdehyde; T-SOD, total superoxide dismutase; CAT, catalase; ALT, alanine aminotransferase; AST, aspartate aminotransferase.

**Figure 3 animals-12-03218-f003:**
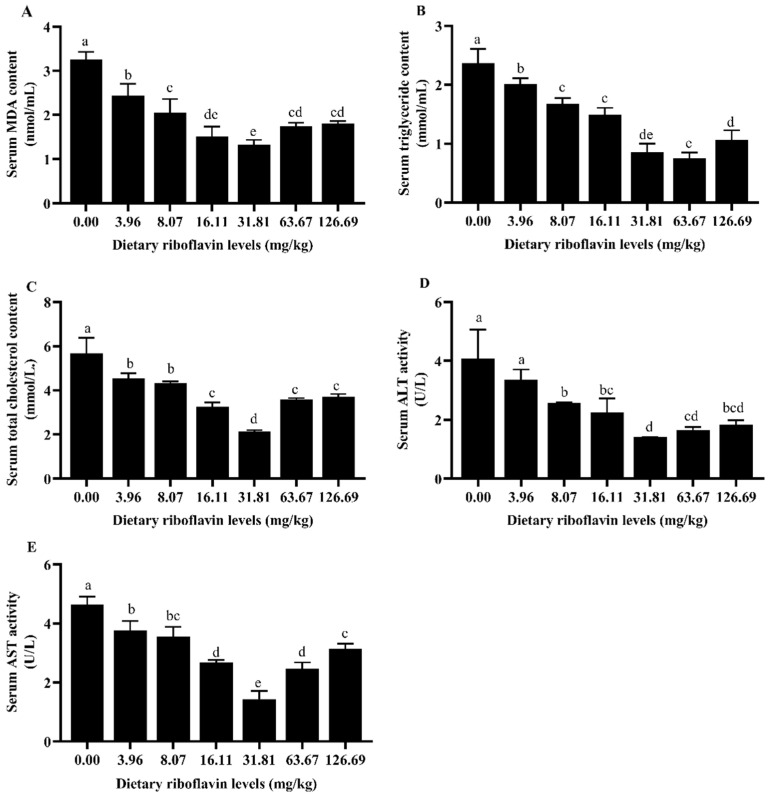
Effect of graded dietary riboflavin levels on serum MDA, triglyceride, total cholesterol, ALT and AST activities of coho salmon (*Oncorhynchus kisutch*) post-smolts for 12 weeks. (**A**) MDA; (**B**) triglycerides; (**C**) total cholesterol; (**D**) ALT; (**E**) AST. Values are expressed as the mean ± SD of three replicate groups. Means with different superscript letters are significantly different (*p* < 0.05). Abbreviations: MDA, malondialdehyde; ALT, alanine aminotransferase; AST, aspartate aminotransferase.

**Table 1 animals-12-03218-t001:** Formulation and proximate composition of the experimental diets for coho salmon (*Oncorhynchus kisutch*) post-smolts (% dry matter).

Ingredients	Dietary Riboflavin Level						
	0.00	3.96	8.07	16.11	31.81	63.67	126.69
Casein ^1^	38.0	38.0	38.0	38.0	38.0	38.0	38.0
Gelatin ^1^	12.0	12.0	12.0	12.0	12.0	12.0	12.0
Corn oil ^1^	6.0	6.0	6.0	6.0	6.0	6.0	6.0
Fish oil ^1^	3.0	3.0	3.0	3.0	3.0	3.0	3.0
Dextrin ^1^	28.0	28.0	28.0	28.0	28.0	28.0	28.0
α-cellulose ^1^	8.0	8.0	8.0	8.0	8.0	8.0	8.0
Mono-calcium phosphate ^1^	1.5	1.5	1.5	1.5	1.5	1.5	1.5
Mineral premix ^2^	2.5	2.5	2.5	2.5	2.5	2.5	2.5
Vitamin premix (riboflavin free) ^3^	1.0	1.0	1.0	1.0	1.0	1.0	1.0
Riboflavin (mg/kg)	0.0	4.0	8.0	16.0	32.0	64.0	128.0
Proximate composition							
Moisture (%)	11.23	11.26	11.25	11.24	11.04	10.80	10.88
Crude protein (%)	46.36	45.67	45.79	46.15	46.45	46.11	46.02
Crude lipid (%)	9.93	9.44	9.51	9.62	9.43	9.47	9.55
Ash (%)	5.44	5.27	5.55	5.42	5.70	5.92	5.37
Gross energy (MJ/kg)	21.07	21.23	21.45	21.53	21.29	21.40	21.22
Riboflavin (mg/kg)	0.00	3.96	8.07	16.11	31.81	63.67	126.69

^1^ Provided by Shandong Conqueren Marine Technology Co., Ltd., Weifang, China. ^2^ Composition (g/kg mineral premix): AlK(SO_4_)_2_·12H_2_O, 123.7; CuSO_4_.5H_2_O, 32.0; CoCl_2_·6H_2_O,49.0; FeSO_4_.7H_2_O, 707.0; MgSO_4_.7H_2_O, 4317.0; MnSO_4_.4H_2_O, 31.0; KI, 5.3; NaCl, 4934.0; Na_2_SeO_3_.H_2_O, 3.4; ZnSO_4_.7H_2_O,177.0. ^3^ Vitamin premix supplied the diets with (mg/kg dry diet): cholecalciferol, 0.04; α-tocopherol, 50; menadione, 44.0; thiamine-HCl, 12.0; D-calcium pantothenate, 20.0; pyridoxine-HCl, 15.0; choline chloride, 500.0; meso-inositol, 200.0; D-biotin, 0.5; folic acid, 1.5; ascorbic acid, 100.0; niacin, 75.0; cyanocobalamin, 0.01.

**Table 2 animals-12-03218-t002:** Effects of graded dietary riboflavin levels on survival, growth performance and feed utilization of coho salmon (*Oncorhynchus kisutch*) post-smolts for 12 weeks.

Dietary Riboflavin Levels (mg/kg)	0.00	3.96	8.07	16.11	31.81	63.67	126.69	*p* Value
Survival rate (%)	96.67 ± 3.33	96.67 ± 3.33	100.00 ± 0.00	96.67 ± 3.33	100.00 ± 0.00	100.00 ± 0.00	96.67 ± 3.33	0.798
Initial body weight (g)	184.12 ± 1.47	185.27 ± 1.02	187.67 ± 0.24	185.68 ± 1.75	187.58 ± 0.55	187.22 ± 1.13	185.99 ± 1.82	0.480
Final body weight (g)	480.28 ± 3.29 ^d^	490.13 ± 1.94 ^cd^	517.11 ± 5.78 ^b^	520.36 ± 4.66 ^b^	537.54 ± 2.33 ^a^	521.05 ± 4.59 ^b^	494.7 ± 1.34 ^c^	<0.001
WGR	160.91 ± 3.72 ^c^	164.55 ± 0.9 ^c^	175.55 ± 3.3 ^b^	180.33 ± 5.1 ^ab^	186.58 ± 2.06 ^a^	178.31 ± 2.32 ^ab^	166.03 ± 2.3 ^c^	<0.001
Food intake	773.3 ± 6.16	778.15 ± 4.27	788.21 ± 1.01	779.87 ± 7.36	787.82 ± 2.3	786.34 ± 4.74	781.16 ± 7.65	0.408
SGR (%/day)	1.14 ± 0.02 ^c^	1.16 ± 0.01 ^c^	1.21 ± 0.02 ^b^	1.23 ± 0.03 ^ab^	1.25 ± 0.01 ^a^	1.22 ± 0.01 ^ab^	1.17 ± 0.01 ^c^	<0.001
FCR	2.18 ± 0.05 ^a^	2.13 ± 0.01 ^a^	2.00 ± 0.04 ^b^	1.94 ± 0.06 ^bc^	1.88 ± 0.02 ^c^	1.96 ± 0.03 ^bc^	2.11 ± 0.03 ^a^	<0.001
CF (g/cm^3^)	1.68 ± 0.04	1.68 ± 0.02	1.60 ± 0.04	1.63 ± 0.07	1.66 ± 0.01	1.62 ± 0.01	1.64 ± 0.02	0.053
HSI (%)	0.95 ± 0.03	1.03 ± 0.04	0.99 ± 0.04	0.93 ± 0.04	0.99 ± 0.03	1.05 ± 0.04	0.96 ± 0.03	0.658
VSI (%)	7.81 ± 0.45	8.02 ± 0.33	8.35 ± 0.52	7.96 ± 0.42	8.15 ± 0.17	8.43 ± 0.15	8.23 ± 0.18	0.615

Values are presented as mean ± SD of three replicate groups. Means in the same row with different superscript letters are significantly different (*p* < 0.05).

**Table 3 animals-12-03218-t003:** Effects of dietary riboflavin levels on the muscle proximate compositions of coho salmon (*Oncorhynchus kisutch*) post-smolts for 12 weeks.

Dietary Riboflavin Levels (mg/kg)	Moisture (%)	Crude Protein (%)	Crude Lipid (%)	Ash (%)
0.00	74.33 ± 0.51	20.32 ± 0.23	3.29 ± 0.12 ^c^	2.60 ± 0.19
3.96	74.76 ± 0.11	19.85 ± 0.28	3.44 ± 0.18 ^bc^	2.60 ± 0.10
8.07	75.34 ± 0.26	20.05 ± 0.31	3.55 ± 0.11 ^b^	2.40 ± 0.12
16.11	75.05 ± 0.14	20.32 ± 0.42	3.66 ± 0.04 ^ab^	2.52 ± 0.25
31.81	74.25 ± 0.43	20.39 ± 0.29	3.81 ± 0.06 ^a^	2.65 ± 0.15
63.67	75.67 ± 0.79	20.37 ± 0.35	3.74 ± 0.12 ^ab^	2.32 ± 0.18
126.69	74.33 ± 0.36	20.25 ± 0.43	3.65 ± 0.07 ^ab^	2.68 ± 0.26
*p* value	0.198	0.625	0.049	0.433

Values are presented as mean ± SD of three replicate groups. Means in the same column with different superscript letters are significantly different (*p* < 0.05).

**Table 4 animals-12-03218-t004:** Measures of the biochemical indicators of liver riboflavin status at the conclusion of the 12-week feeding trial.

Dietary Riboflavin Levels (mg/kg)	Liver Riboflavin (mg/kg)
0.00	5.09 ± 0.1 ^d^
3.96	11.84 ± 0.12 ^c^
8.07	12.52 ± 0.33 ^b^
16.11	13.89 ± 0.06 ^a^
31.81	14.06 ± 0.11 ^a^
63.67	13.88 ± 0.08 ^a^
126.69	13.79 ± 0.06 ^a^
*p* value	< 0.01

Values are presented as mean ± SD of three replicate groups. Means in the same column with different superscript letters are significantly different (*p* < 0.05).

## Data Availability

All the data in the article are available from the corresponding author upon reasonable request.

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
