# Peer review of "Effects of Dietary Riboflavin Supplementation on the Growth Performance, Body Composition and Anti-Oxidative Capacity of Coho Salmon (Oncorhynchus kisutch) Post-Smolts"

_animals, 2022, doi:10.3390/ani12223218_

Round 1
Reviewer 1 Report
This study investigated the effects of dietary riboflavin supplementation on the growth performance, body composition and anti-oxidative capacity of coho salmon (Oncorhynchus kisutch) post-smolts", providing a reference for better understanding the dietary riboflavin requirement for this species.
The manuscript could be considered for publication after being minor revisions. Some information needs to be provided as follows.
Specific comments:
1. Line 39: p < .05 should be revised to p < 0.05. Similar results should be described in detail.
2. Line 101: Recommended to add “ The purified basal diet was formulated using casein as a protein source and fish oil and corn oil as lipid sources.” after the first sentence.
3. Table 1 “Dietary linoleic acid level”should be “Dietary riboflavin levels”? Please check it carefully. And please provide the purity of riboflavin.
4. Line 123-126: According to the description, the mean initial weight of 186.22 ± 0.41 g looks like an average weight of the total fish used in the beginning of the trial rather than the initial weight of each treatment. If this is the case, I would like to know the method to obtain the identical fish initial body weight of 186.22 ± 0.41 g for each triplicate treatment.
5. Were the fish placed in each tank bulk-weighed immediately before the start of the trial to obtain information about the biomass present in each tank? At the end of the trial were the fish in each tank weighed individually or in bulk?
6. Line 128: “14.0 to 16.0 ℃” should be changed to “14 to 16 ℃”, consistent with the following text.
7. Line 130: What does natural light regimes mean? Were the fish reared outdoor?
8. Line 136: Were the livers from the 5 fish sampled from each tank pooled prior to homogenization and analysis, giving a single liver homogenate for each tank?
Reviewer 2 Report
The present manuscript certainly presents an interest for the scientific community since the authors report the positive effects on an enriched diet with riboflavin on the body composition an anti-oxidative capacity.
- The viscerosomatic index refers to the weight of the viscera package with respect to body weight and in you it refers to the weight of the intestine with respect to body weight. Please explain what is the weight used to make the index.
-Line 171-174: refers to the fact that there are no significant differences between the indices for any quantity, however later on it does say that there is a difference. Explain this better.
- Check the legends of the tables and figures, make sure you follow the author's guide.
- The journals must have the abbreviations of the names to follow the guide of authors.
Reviewer 3 Report
Comments to the Author
General comments:
The manuscript “the effects of dietary riboflavin supplementation on the growth performance, body composition and anti-oxidative capacity of coho salmon (Oncorhynchus kisutch) post-smolts” focused on the vitamin requirement of coho salmon nutrition, which is essential for the production of high quality and profitability for aquaculture farmers.
The manuscript has merit but requires minor revision before it can be accepted for publication in the Journal of Animals.
Specific comments:
1. Please check all tables, the data in the table should be aligned.
2. Line 123-126: Were the fish placed in each tank bulk-weighed immediately before the start of the trial to obtain information about the biomass present in each tank? This is not mentioned specifically in the section entitled 'Feeding trial'.
3. Whether the floating cages of different experimental groups are randomly distributed?
4. Line 132-143: Were feed deprived for fish before the biometrics and sampling?
5. Line 150: How long is muscle moisture measured?
6. In table 2: The unit for SGR should be , not %/d, consistent with above; the unit for CF is “g / cm3” or “%”, please determine it.
7. In figure 1: The unit of SGR should be , please correct it.
8. Make sure that symbols, sub- and super-scripts, upper- and lower-case are presented correctly, and that there is correct and consistent use of italics, brackets and punctuation etc.
Reviewer 4 Report
This study investigated the optimal riboflavin requirement of coho salmon post-smolts based on growth performance and antioxidant capability. Generally, the manuscript was arranged in a straight-forward way, and is easy to understand. The experimental design is scientifically sound. The results obtained could advance the development of high efficiency feed for this species. However, there are some major concerns that should be addressed by the authors. Please refer to the following comments.
1. Materials and methods: please provide the reference numbers of the commercial kits, and the analyzing principles they follow (lines 155-161).
2. Data analysis: 1) Were the data tested for homogeneity and normality before the conduction of ANOVA? 2) One-way ANOVA and mean separation is not an appropriate analysis for a quantitative independent variable such as graded levels of riboflavin. A more appropriate analysis is to analyze all the responses using polynomial orthogonal contrasts so you can indicate whether there are significant linear, quadratic or cubic responses. 3) The second-order polynomial regression analysis might be more appropriate to determine the regression of SGR against dietary riboflavin levels, since SGR decreased significantly as dietary riboflavin levels increased from 31.81 to 126.96 mg/kg.
3. Results: several key parameters were missing, including weight gain, feed consumption and liver riboflavin content.
4. Please justify the analysis of proximate composition in muscle (Table 3). Why not use whole body?
5. Please justify the measurement of hepatic ALT and AST activities (Fig. 2), which were generally related to amino acid metabolism. The discussion of them is also missing in the discussion section.
6. The description of MDA, total cholesterol and triglyceride were wrong (Fig. 2 and 3, lines 196-211). They are not enzymes, so the word “activity” is not appropriate. This also held true for the use of “content” for both ALT and AST, which are indeed enzymes.
7. The discussion section is quite descriptive with the present results poorly discussed. The authors should provide more explanations for the key findings with the underlying mechanisms interpreted.
Round 2
Reviewer 4 Report
The authors have made substantial efforts to revise the manuscript. All the comments have been responded appropriately with the manuscript revised accordingly. The manuscript can be accepted for publication.